# Decreased Memory Suppression Ability in Restrained Eaters on Food Information—Evidence from ERP Experiment

**DOI:** 10.3390/nu17152523

**Published:** 2025-07-31

**Authors:** Qi Qi, Ke Cui, Li Luo, Yong Liu, Jia Zhao

**Affiliations:** 1School of Psychology, Hainan Normal University, Haikou 571158, China; psyqiqi@hainnu.edu.cn; 2Adolescent Psychological Development and Education Center of Hainan, Hainan Normal University, Haikou 571158, China; 3Institute of Psychology, Chinese Academic of Sciences, Beijing 100101, China; trick075@163.com; 4Department of Education, Neijiang Normal University, Neijiang 641100, China; luoliv9@aliyun.com; 5Faculty of Psychology, Southwest University, Chongqing 400715, China; liuy0768@swu.edu.cn; 6Key Laboratory of Cognition and Personality (Ministry of Education), Southwest University, Chongqing 400715, China

**Keywords:** electroencephalogram, active forgetting, excessive dieting behavior, think/no-think paradigm

## Abstract

**Background/Objectives:** Food-related memory influences appetite regulation, with memory inhibition potentially reducing cravings. While obesity is linked to inhibitory deficits, how restrained eating affects memory suppression in healthy-weight individuals remains unclear. This study examined the cognitive and neural mechanisms of food-memory suppression in young women. **Methods:** Forty-two female participants completed a think/no-think task with high-/low-calorie food cues while an EEG was recorded. Event-related potentials (ERPs) were assessed and time–frequency analyses (theta/beta oscillations) were performed. **Results:** Restrained eaters showed reduced memory control for both food types. The ERP analysis revealed significant N200 amplitude differences between think/no-think conditions (*p* = 0.03) and a significant interaction between food calories and think/no-think conditions (*p* = 0.032). Theta oscillations differed by group, food calories, and conditions (*p* = 0.038), while beta oscillations reflected food-cue processing variations. **Conclusions:** In conclusion, restrained eaters exhibit distinct neural processing and attenuated food-memory suppression. These results elucidate the neurocognitive mechanisms underlying dietary behavior, suggesting that targeted interventions for maladaptive eating could strengthen memory inhibition.

## 1. Introduction

Restrained eating—a cognitive behavioral strategy to regulate food intake for weight management—has gained significant prevalence among healthy-weight women in China, propelled by sociocultural ideals that equate thinness with attractiveness [1,2,3,4]. Neural plasticity models propose that sustained dietary restriction may reconfigure neural responses to food cues, particularly within circuits governing attentional bias and cognitive control [5,6,7,8]. Paradoxically, despite its widespread adoption, restrained eating often fails to achieve long-term weight goals [9,10], with lapses frequently attributed to heightened reactivity to palatable foods and deficits in inhibitory control [11,12]. While obesity research documents broad executive dysfunction that disrupts dietary regulation [13,14], restrained eaters typically exhibit context-specific cognitive vulnerabilities limited to food-related processing, rather than generalized impairment [11]. This distinction underscores the necessity to investigate the specialized cognitive mechanisms—particularly memory suppression—within this population.

Memory suppression represents a distinct neurocognitive process that cannot be reduced to general executive function. Paradigms like directed forgetting (‘white bear’ paradigm), retrieval-induced forgetting, and the think/no-think (TNT) task elucidate unique intentional forgetting mechanisms [15,16,17]. Whereas Wegner’s classic experiment demonstrated how suppression attempts paradoxically intensify unwanted thoughts [15], the TNT paradigm operationalizes active memory inhibition through reduced recall of “no-think” items [16]. Critically, food-specific memory control has clinical relevance: restrained eaters suppressing chocolate thoughts subsequently consume larger quantities compared to control groups [18], suggesting maladaptive rebound effects. However, the existing TNT findings remain inconsistent. Zhang et al. reported reduced recall accuracy for suppressed food stimuli [19], but their inclusion of obese participants introduced confounding effects, as visceral adiposity independently compromises memory suppression [20]. Conversely, Bian et al. observed higher recall in no-think versus think conditions [21], though ceiling effects (>90% accuracy) limited the results’ interpretability. These discrepancies highlight methodological constraints and the need to isolate trait-specific effects in homogeneous samples.

Significant gaps persist in current approaches. First, the predominant use of verbal stimuli overlooks fundamental differences in how visual food cues engage cognitive and emotional systems. Food images elicit stronger attentional and affective responses than words, better approximating real-world dietary challenges. Second, while the calorie content differentially modulates cognitive processing [22], its role in memory suppression remains unexplored. Third, electrophysiological investigations have narrowly focused on event-related potentials (e.g., N200 indexing conflict monitoring [20,21]), neglecting the oscillatory dynamics specifically implicated in memory control.

An electroencephalogram (EEG) offers powerful insights through neural oscillations [23].The N200 component is related to cognitive control and conflict monitoring [20,21]. Theta (4–8 Hz) and beta rhythms coordinate neural activities during cognitive control [24,25,26,27] and memory suppression [27,28,29]. Though these signatures are well-established in executive control research [30,31], their role in food-specific memory suppression among restrained eaters remains unexamined—a notable omission given evidence that theta/beta dynamics successfully decode food-related inhibition in a clinical population [27].

In summary, the aim of this study was to explore whether food suppression ability was changed in restrained eaters due to their daily dieting behavior with a high-ecological-validity TNT paradigm (high-/low-calorie food pictures). We hypothesized that (1) behaviorally, restrained eaters would exhibit decreased memory suppression for high-calorie foods (higher recall rates) when compared to low-calorie foods, whereas controls would show no calorie-based differences; (2) neurally, restrained eaters would display enhanced early attentional processing to food stimuli, greater conflict monitoring during suppression attempts, and dysregulated theta and beta oscillations reflecting inefficient top-down control over food-related memories.

## 2. Materials and Methods

### 2.1. Participants

A total of 144 potential participants were initially recruited for this study. Subsequent screening using the Dutch Eating Behavior Questionnaire—Restraint Scale (DEBQ-RS) established eligibility. From this screened cohort, a final sample of 42 female participants was randomly selected and classified into two groups based on their scores on the DEBQ-RS, the normal group (n = 21, DEBQ-RS < 3) and the restrained eater group (n = 21, DEBQ-RS ≥ 3), according to the previous studies [20,32,33]. It should be stated that only women were recruited because empirical evidence indicates gender-specific approaches to weight management: men predominantly utilize exercise for weight control, while women overwhelmingly rely on dietary restraint due to stronger body image concerns [2]. The participants’ demographic information is summarized in Table 1. All participants had healthy body mass indices (BMIs) between 18.5 and 23.9 kg/m^2^ and normal/corrected-to-normal vision. The participants had no recorded history of neurological or psychiatric illnesses and had no eating disorders other than restrained eating behaviors with the amount of high-calorie food reduced. They confirmed their participation in the experiment and signed an informed consent file, and they received CNY 60 as compensation for their participation after the experiment. The local Ethics Committee approved the study protocol, and all procedures followed the Declaration of Helsinki.

### 2.2. Materials

#### 2.2.1. Self-Measurements

The DEBQ, a validated questionnaire with 33 items, evaluates eating behaviors through three subscales: emotional, external, and restrained eating [34]. This study specifically utilized the Restraint Scale (DEBQ-RS), a 10-item subscale (e.g., “Do you take into account your weight when you eat?” and “When your weight increases, do you eat less than usual?”), to assess dietary restraint. A 5-point Likert scale was utilized to rate each item from (1) to (5), indicating “never” to “always”, with greater scores indicating greater dietary restraint and control over eating habits. The internal consistency of the DEBQ-RS is consistently high (Cronbach’s α = 0.92–0.94) regardless of weight classification in nonclinical samples of healthy-weight, overweight, and obese individuals [34]. It has been translated into Chinese and widely applied in prior research [27,35,36].

Participants with mean DEBQ-RS scores greater than or equal to 3 were categorized as restrained eaters, while those with mean scores below 3 served as controls. Due to the task complexity of the think/no-think paradigm, only the age, BMI, and DEBQ-RS scores were recorded from the participants to reduce the respondent burden and attrition with an increase in survey length.

#### 2.2.2. Experimental Procedure

The think/no-think (TNT) task was employed here, consisting of three sequential phases: learning, think/no-think, and testing (Figure 1).

 **(1)** 
**Learning phase**


During the learning phase, the participants were instructed to memorize 60 picture pairs, each consisting of one neutral cue image (selected from the International Affective Picture System, IASP) and one target food image (from a food picture gallery [37]). The 60 pairs were evenly divided into two blocks: 30 pairs featuring food cues with high calorie contents and 30 pairs featuring food cues with low calorie contents. The calorie information for the food pictures was assessed by two bromatology professors from Southwest University and further rated by a sample comprising 989 students from secondary schools (junior and senior high) and universities in China [37]. These pairs were presented in two separate blocks, with all pairs from one calorie category shown before moving to the next. After memorizing the first set of 30 picture pairs, the participants completed a recall test. During this test, the participants were shown each cue picture and asked to name aloud the corresponding food image. Participants who failed to recall at least 60% of the pairs were given a second opportunity to review and memorize the same set of picture pairs. Each picture pair was displayed for 4 s in a randomized sequence during the learning phase. The experimenter recorded the participants’ responses during the recall test, categorizing and calculating accuracy based on the assignment of picture pairs to think/no-think conditions and food types.

 **(2)** 
**Think/No-think phase**


If participants achieved an accuracy rate of at least 60% in the learning phase, they advanced to the second phase: the think/no-think phase. In this phase, each trial displayed a cue picture surrounded by either a red or green square for 4 s. A red square indicated that participants should actively suppress the memory of the target picture (no-think condition), while a green square instructed them to recall the target picture (think condition). This red/green setting for the conditions was adopted from previous studies where red-colored words were set for the no-think condition, while green-colored words were set for the think condition [19,20,21]. The ratio of think trials to no-think trials was 2:1.

 **(3)** 
**Test phase**


After completing the TNT phase, the participants entered the test phase. During this phase, the red and green squares were removed from the cue pictures. The participants were shown all 60 cue pictures and were required to verbally recall the name of the corresponding food target image. The recall accuracy was calculated for both the think and no-think conditions, as well as for high- and low-calorie food types.

All experimental trials were controlled and presented via E-prime 2.0 software. A total of 60 neutral cue images were selected from the IASP, paired with 60 food target images—30 high-calorie and 30 low-calorie food pictures.

#### 2.2.3. Data Recording and Analysis

(1)Behavior analyses

The mean (M) and standard deviation (SD) of the self-reported variables, including age, DEBQ-RS scores, and BMI, were calculated for each group (Table 1). Independent-sample *t*-tests were utilized to inspect group differences in self-reported measures.

For task performance, the accuracy (ACC) data for the think/no-think conditions, high-/low-calorie food categories, and participant groups (restrained eaters and normal controls) were calculated based on the experimenter’s recording (Table 2). A 2 × 2 × 2 × 2 repeated-measures analysis of variance (ANOVA) was performed to analyze these data, with group (restrained eaters vs. normal controls) as a between-subject factor and condition (think vs. no-think), food type (high-calorie vs. low-calorie), and test time (pre-test vs. post-test) as within-subject factors. All statistical analyses were conducted using SPSS software version 27 (IBM, Armonk, NY, USA).

(2)EEG Recording and analyses

NeuroScan SynAmps2 amplifiers (Compumedics NeuroScan, Charlotte, NC, USA) were used to record EEGs from 64 Ag/AgCl electrodes positioned according to the [10-20/10-10] international system with a 1000 Hz sampling rate. Electrodes M1 and M2 were placed on the left and right mastoids, serving as reference channels. The participants were required to reduce their movement throughout the experiment to ensure high-quality EEG recordings.

The EEG data preprocessing and analysis were performed using EEGLAB v2022.0 (Swartz Center for Computational Neuroscience, La Jolla, CA, USA) [38], an open-source MATLAB R2024a toolbox (The MathWorks, Inc., Natick, MA, USA). The data were preprocessed using a finite impulse response (FIR) bandpass filter with a range of 0.1–30 Hz. EEG trials were then extracted within a time window of −200 ms to 800 ms relative to stimulus onset and visually inspected on a trial-by-trial basis. Trials containing excessive amplitudes or chaotic fluctuations were excluded. Artifact correction was performed using independent component analysis (ICA), which decomposed the EEG data into independent components. Components corresponding to ocular movements (identified via topographic maps and time-course characteristics), muscular artifacts, and noisy channels were manually rejected [35,39]. After artifact removal, the cleaned trials were re-examined to make sure that no artifacts or invalid data remained [40]. In total, the statistical distribution for the number of removed trials was 0.85 ± 2.75 (mean ± std), and the number of removed independent components was 2.7 ± 1.4. The EEG data for two participants were removed due to multiple slow fluctuations caused by sweat. The grand-averaged ERP waveforms of the remaining participants were computed for each group (restrained eaters and controls), using a pre-stimulus baseline of −200 ms to 0 ms. Based on prior research and the experimental design, the N200 ERP component (230–260 ms) was analyzed based on the average from selected electrodes (F3/Fz/F4) at the frontal cortex. These components were compared across the participant groups and food types (high- versus low-calorie).

A time–frequency analysis (TFA) was conducted using the short-time Fourier transformation (STFT) algorithm, where optimization parameters were set (a 200 ms window size and a 50 ms step size) to balance the time and frequency resolutions [39,41]. Baseline normalization was performed on the TFA results using the pre-stimulus period (−200 ms to 0 ms) as a reference. The percentage baseline correction method was employed, where the corrected power (p_corrected_) was calculated as(1)$$p_{corrected}=(p-p_{0})/p_{0}$$
Here, *p* represents the measured amplitude power, and *p*_0_ is the mean baseline power. The grand-averaged time–frequency (TF) results were computed for each electrode under different conditions and food types. Regions of interest (ROIs) were identified for the theta (4–8 Hz, 0.2–0.8 s) and beta (13–20 Hz, 0.2–0.8 s) bands of three electrodes (F3/Fz/F4) in the frontal cortex and were compared across conditions.

Statistical analyses were performed using SPSS version 27 (IBM, Armonk, NY, USA). Repeated-measures ANOVAs were conducted on the mean amplitudes of N200, theta power, and beta power. Each ANOVA featured a 2 × 2× 2 × 2 × 2 design: group (REs versus controls), food type (high- versus low-calorie), condition (think versus no-think), and measurement time (recall rate during learning versus test phase). Simple effects analyses were conducted as needed, based on the ANOVA results.

## 3. Results

### 3.1. Self-Reported Results

A repeated-measures ANOVA (RMANOVA) with factors of group (restrained eaters vs. controls), food type (high-calorie vs. low-calorie), condition (think vs. no-think), and test phase (pre-test during learning phase vs. post-test during test phase) revealed significant main effects for high- vs. low-calorie foods (F(1,40) = 19.456, *p* < 0.001, η^2^ = 0.327) and think vs. no-think (F(1,40) = 8.458, *p* = 0.006, η^2^ = 0.175); a significant interaction effect between pre-/post-tests and groups (F(1,40) = 4.952, *p* = 0.032, η^2^ = 0.110); and a significant interaction effect between high-/low-calorie foods and think/no-think stimuli (F(1,40) = 8.297, *p* = 0.006, η^2^ = 0.172).

It can be found from Table 1 that ACC was higher for the think condition than for the no-think condition, and ACC was higher for low-calorie food than for high-calorie food. The simple effects analysis showed that ACC for low-calorie food was significantly higher than that for high-calorie food under the no-think condition (*p* < 0.001), and ACC was higher for think trials than for no-think trials under high-calorie food stimuli (*p* < 0.001). Furthermore, although the simple effects analysis for pre-/post-tests and groups was not significant, the normal group displayed decreased ACC from pre- to post-test (*p* = 0.096), while the RE group displayed increased ACC from pre- to post-test (*p* = 0.157).

### 3.2. EEG Results

The ERP fluctuations for the two groups under think/no-think conditions and high-/low-calorie food stimuli are shown in Figure 2.

(1)ERP results

Significant main effects for N200 amplitudes were found between the think and no-think conditions (F(1,38) = 5.086, *p* = 0.030, η^2^ = 0.118), where N200 under the think condition was significantly greater than that under the no-think condition. A significant interaction effect was found for think/no-think conditions and high-/low-calorie food stimuli (F(1,38) = 4.935, *p* = 0.032, η^2^ = 0.115). A simple effects analysis found that the N200 amplitude for low-calorie food stimuli under the think condition was significantly greater than that under the no-think condition. No other significant effect was found.

(2)TFA results

The TFA results for the two groups under think/no-think conditions and high-/low-calorie food stimuli are shown in Figure 3, where the regions of interest (ROIs) including theta and beta power are marked.

The RMANOVA for theta power found a significant interaction effect between groups, think/no-think conditions, and high-/low-calorie food stimuli (F(1,38) = 4.64, *p* = 0.038, η^2^ = 0.109). A simple effect analysis found that the theta power of the control group under the no-think condition was much larger than that under the think condition for low-calorie food (*p* = 0.046), while this effect was not found in the restrained eaters. In addition, the theta power for the restrained eaters under the no-think condition was much higher (marginally significant) than that under the think condition for high-calorie food stimuli (*p* = 0.097), which was not identified in the control group.

The RMANOVA for beta power found a significant main effect between high- and low-calorie food stimuli (F(1,38) = 7.761, *p* = 0.008, η^2^ = 0.17). It was found that the beta of high-calorie food was much more negative than that of low-calorie food. No other significant effect was found for beta power.

## 4. Discussion

This study investigated memory suppression in restrained eaters (REs) using a think/no-think (TNT) paradigm with high- and low-calorie food stimuli. Our core findings revealed three key phenomena: First, the recall accuracy was significantly higher for think versus no-think trials, consistent with the TNT paradigm’s established efficacy in modulating food-related memories [19,20]. Second, low-calorie foods exhibited lower forgetting compared to high-calorie foods, indicating that the caloric content differentially modulates memory control. Third, and most critically, opposing recall trajectories from pre- to post-test for restrained eaters and controls were observed, where the controls showed the expected decline in recall ACC induced by memory suppression, while the restrained eaters exhibited a paradoxical enhancement of recall post-test, particularly for high-calorie food stimuli. In addition, electrophysiological data further elucidated these effects: the N200 component (indexing conflict monitoring) was amplified for low-calorie foods during think trials [20,21], while divergent theta oscillations between the groups signaled distinct neural resource allocation during suppression.

These outcomes align with and extend the existing literature. The TNT paradigm’s behavioral validity in food-related contexts [19,20] and theta’s established role in long-term memory inhibition [42,43,44,45,46] are reaffirmed. However, several discordances merit analysis. While prior studies associated dietary restraint with generalized food memory biases [15], our observed calorie-specific recall enhancement in REs diverges from TNT predictions. This may reflect REs’ maladaptive prioritization of high-calorie memories: behavioral restraint coexists with hypervigilant encoding, potentially perpetuating disordered eating cycles. Similarly, the unexpected forgetting decrease for low-calorie foods contrasts with studies showing efficient suppression of neutral stimuli [20]. We posit that this stems from perceptual load differences: visually homogeneous low-calorie stimuli (e.g., vegetables) imposed greater retrieval demands than distinct high-calorie items. This should also be the reason for the higher N200 amplitude for low-calorie food stimuli under the think condition than under the no-think condition. As food cues with low caloric value were difficult to remember due to their similar appearance, more conflict was encountered by participants during this period, resulting in higher conflict for participants during the think trials for low-calorie foods. Theta/beta dissociation in REs further diverges from the patterns seen in clinical populations [27], suggesting that subclinical groups deploy compensatory neural effort to override food-cue salience.

The implications of these findings are threefold. Clinically, theta and beta oscillations may serve as biomarkers for early identification of memory control deficits in at-risk individuals. Methodologically, image-based paradigms (vs. verbal) better capture food-specific memory dynamics but introduce visual complexity confounders requiring standardization. Theoretically, caloric value appears to hierarchically modulate memory control: the low-calorie decrease links to perceptual load, while the high-calorie enhancement in REs suggests that incentive salience overrides suppression.

Despite these advances, limitations necessitate caution. The DEBQ-RS group separation (cutoff = 3) proved suboptimal; a post hoc analysis using extreme scores (2.5 vs. 3.5) revealed stronger effects, underscoring the need for larger, stratified samples to enhance statistical sensitivity. The external validity is constrained by (1) the limited stimulus generalizability (homogeneous low-calorie images), (2) unmeasured confounders including fasting status and socio-demographic factors (e.g., cultural dietary norms, age ranges), and (3) the non-clinical sample’s lack of eating disorder screening. Although pre-test familiarity showed no group differences, persistent food-type effects suggest unresolved individual preference influences.

Future research should pursue four directions: first, validate neural markers across clinical populations (e.g., obesity, anorexia nervosa) to establish transdiagnostic utility; second, employ personalized food stimuli calibrated to individual preferences; third, integrate metabolic biomarkers (e.g., ghrelin) to quantify hunger-state modulation of suppression efficacy; and fourth, use longitudinal designs to track whether memory control deficits predict dietary relapse. Such work could transform our mechanistic understanding of cognitive vulnerabilities in disordered eating. Finally, this was the first analysis of the dataset; it can also be explored using the source reconstruction method to test the activities of different brain regions, or compared with the results from other related datasets such as those for participants with obesity or anorexia nervosa.

## 5. Conclusions

In summary, our study successfully validated a substantial portion of the hypotheses. The restrained eaters exhibited unique neural and behavioral signatures of memory control failure—particularly for high-calorie foods. Despite this study’s advanced methodology through image-based TNT tasks and electrophysiological dissection of the suppression mechanisms, addressing sampling constraints and ecological validity remains imperative for clinical translation.

## Figures and Tables

**Figure 1 nutrients-17-02523-f001:**
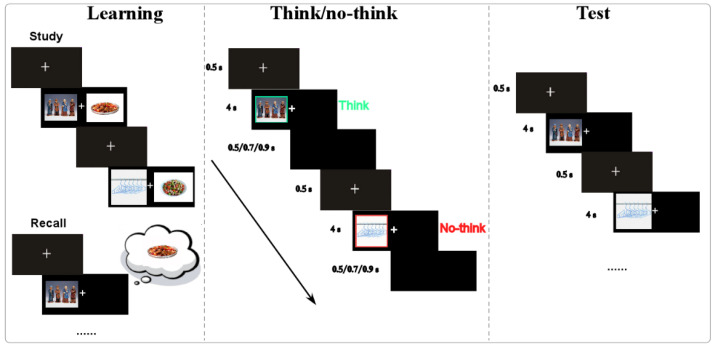
The experimental procedure for the think/no-think (TNT) task. In this study, 60 picture pairs (30 for the think condition, 30 for the no-think condition) were applied. After learning the cue–target associations during the learning phase, the participants proceeded to the TNT phase, where they were instructed to actively recall the target images for think items (marked with green) and to suppress the target images for no-think items (marked with red). In the final test phase, the participants were required to recall the target image when presented with the corresponding cue image, which was displayed without the colored indicators.

**Figure 2 nutrients-17-02523-f002:**
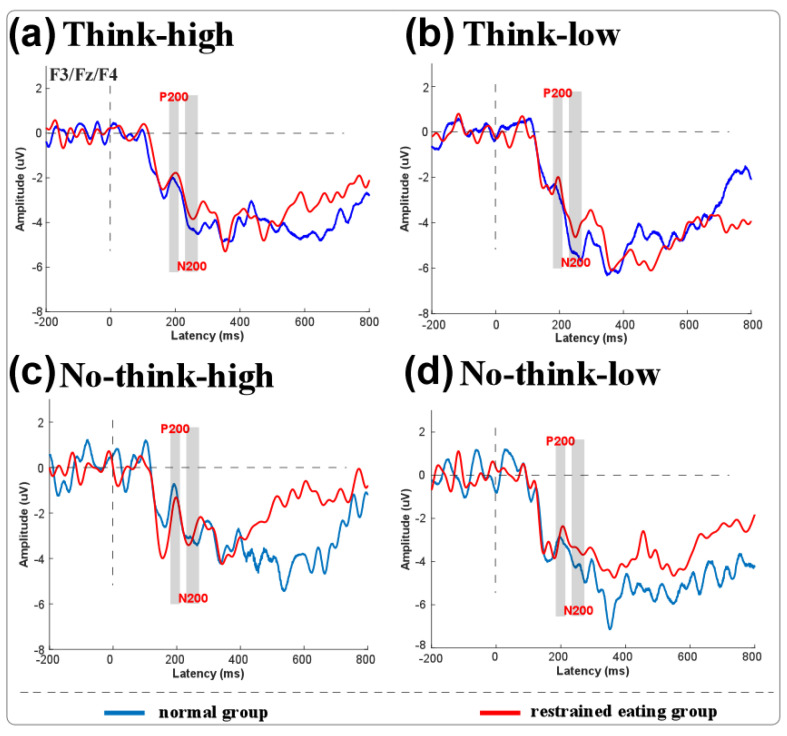
Results for ERP components of normal (blue line) and restrained eaters (red line) under think conditions with high- (**a**) and low-calorie food stimuli (**b**), and under no-think conditions with high- (**c**) and low-calorie food stimuli (**d**). N200 (230–260 ms) component is marked with dark color.

**Figure 3 nutrients-17-02523-f003:**
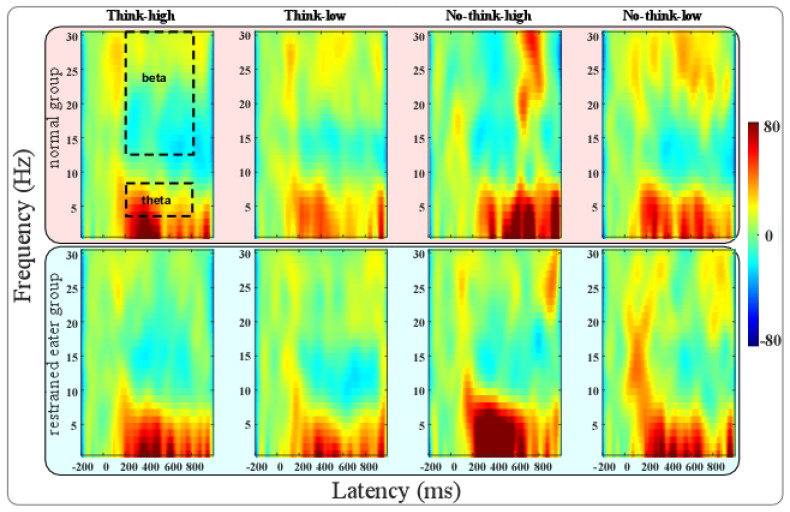
Results for time–frequency analysis (TFA) of normal (upper part) and restrained eaters (lower part) under different conditions. Regions of interest (ROIs) for beta (0.2–0.8 s, 13–30 Hz) and theta (0.2–0.8 s, 4–8 Hz) power are marked in figure.

**Table 1 nutrients-17-02523-t001:** Demographic characteristics and subjective reports.

Variable	Normal Group (M ± SD) N = 21	RE Group (M ± SD) N = 21	*p*
Age *	19.00 (1.10)	19.95 (2.16)	0.079
DEBQ-RS ***	2.36 (0.54)	3.53 (0.40)	<0.001
BMI	21.41 (4.57)	21.32 (1.29)	0.936

Note: * *p* < 0.05; *** *p* < 0.001. DEBQ-RS: Dutch Eating Behavior Questionnaire—Restraint Subscale; BMI: body mass index; RE: restrained eater.

**Table 2 nutrients-17-02523-t002:** Mean recalling ACC for two groups under different conditions.

	Experimental Conditions	Normal Group (M ± SD) N = 21	RE Group (M ± SD) N = 21
Pre-test	Think–Low-calorie food	0.853 (0.024)	0.842 (0.024)
	Think–High-calorie food	0.815 (0.026)	0.803 (0.026)
	No-think–Low-calorie food	0.858 (0.029)	0.843 (0.029)
	No-think–High-calorie food	0.765 (0.037)	0.680 (0.037)
Post-test	Think–Low-calorie food	0.842 (0.025)	0.863 (0.025)
	Think–High-calorie food	0.790 (0.026)	0.835 (0.026)
	No-think–Low-calorie food	0.828 (0.036)	0.850 (0.036)
	No-think–High-calorie food	0.720 (0.043)	0.714 (0.043)

## Data Availability

The data presented in this study are available on request from the corresponding author. The data are not publicly available due to concerns about privacy and ethics in personal decision-making.

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
