# Peer review of "Decreased Memory Suppression Ability in Restrained Eaters on Food Information—Evidence from ERP Experiment"

_nutrients, 2025, doi:10.3390/nu17152523_

Round 1

Reviewer 1 Report

Comments and Suggestions for Authors
  • The design of the study needs to be mentioned in the title, cross-sectional? Longitudinal? Observational? Experimental?
  • The keywords should differ from those, which appear in the title, to increase the article reachability once published.
  • The abstract should be structured in subsections as MDPI guidelines: Background & Aim; Methods; Results and Conclusion.
  • The abstract section should be shortened and focused to appear more concise. For instance the Background & Aim subsection should not exceed a couple of statements. The Method subsection should include the sample size. The Result subsection should include numerical results. The Conclusion subsection should include the practical potential implication.
  • The end of the Introduction section should state the aim(s) stated clearly, followed by an expected hypothesis.
  • In the Method section authors should specify how the sample size was determined, they recruited 42 subject and by chance were equally distributed in n=21 as normal and n=21 in the restrained eater group, or it has been done a sort of selection?
  • The socio-demographic except age are lacking, why? This is an important limitation.
  • Authors should clearly state if the original version of DEBQ-RS, or a Chinese validated one, if so, they have to cite the publication of Chinese validated version.
  • Authors should state how many publications are related to the same dataset.
  • It is not clear if authors tested the distribution of the sample and checked its normality. This should be reported. As if the sample is not normally distributed, means cannot be used, neither the ANOVA for repeated measures.      
  • In the Discussion section initially authors should mention their main findings, and extensively report a sort of comparison in concordance and discordance with their finding, as well as the potential reason behind.
  • In the Discussion section in a second place authors should mention the main implication of their finding. Followed by stating clearly and extensively the strength and limitations of their study, foremost the sample size, the external validity.
  • In the Discussion section authors should conclude with the needed new directions for future on the topic.

Author Response

Responds to the comments of Reviewer 1:

Reviewer 1:

  1. The design of the study needs to be mentioned in the title, cross-sectional? Longitudinal? Observational? Experimental?

Response: Thank you very much for the comment. It was an experimental study with EEG technique, and we have revised the title of manuscript according to the comments, marked with blue color as “Decreased memory suppression ability in restrained eaters on food information-Evidence from ERP experiment”.

  1. The keywords should differ from those, which appear in the title, to increase the article reachability once published.

Response: Thank you for your comments. We have revised the keywords to avoid repetition to the title.

  1. The abstract should be structured in subsections as MDPI guidelines: Background & Aim; Methods; Results and Conclusion. The abstract section should be shortened and focused to appear more concise. For instance the Background & Aim subsection should not exceed a couple of statements. The method subsection should include the sample size. The result subsection should include numerical results. The conclusion subsection should include the practical potential implication.

Response: Thank you very much for your comments. We have rewritten the abstract in the revised manuscript according to your comments.

  1. The end of the Introduction section should state the aim(s) stated clearly, followed by an expected hypothesis.

Response: Thank you very much for your comments. We have added the aim of study at the end of introduction in the revised manuscript.

  1. In the method section authors should specify how the sample size was determined, they recruited 42 subject and by chance were equality distributed in n=21 as normal and n=21 in the restrained eater group, or it has been done a sort of selection?

Response: Thank you for your comments. A total of 144 potential participants were initially recruited for the study. Subsequent screening using the Dutch Eating Behavior Questionnaire-Restraint Scale (DEBQ-RS) established eligibility. From this screened cohort, a final sample of 42 participants was randomly selected. This sample was stratified to comprise two distinct subgroups: 21 participants exhibiting DEBQ-RS scores > 3 and 21 participants exhibiting DEBQ-RS scores ≤3. This methodological detail regarding participant selection based on the DEBQ-RS threshold has been incorporated into the revised manuscript.

  1. The socio-demographic except age are lacking, why? This is an important limitation.

Response: Thank you for your comments. In our study, we just recorded the essential information with age, BMI, and DEBQ-RS scores, which was referred to one of published papers in our group (Cui et al., 2023, Higher visceral adipose tissue is associated with decreased memory suppression ability on food-related thoughts: A 1-year prospective ERP study. Appetite, 191: 107048.). It should be more rigorous to record more socio-demographic information about the participants. We sincerely appreciate the reviewer’s astute observation regarding the limited socio-demographic data in our study. This is indeed a recognized limitation, and we address it as follows:

  1. Methodological Rationale:

During participant recruitment, our primary focus was on stratifying the cohort based on DEBQ-RS scores and age due to their direct relevance to our core research question examining age- and restraint-specific eating behaviors. Collecting extensive socio-demographic variables (e.g., income, education, ethnicity) was constrained by:

Survey length restrictions to reduce respondent burden and attrition as the think/no-think paradigm was much difficult with large burden at memory and attention.

  1. Acknowledgment of Limitation:

We fully concur that this omission restricts the generalizability of our findings and obscures potential socio-demographic confounders. This limitation has been explicitly added to the ‘Study Limitations’ section of the revised manuscript:

"The absence of comprehensive socio-demographic data (e.g., socioeconomic status, cultural background) limits contextual interpretation of results and warrants caution in generalizing findings beyond the studied age-restraint subgroups."

  1. Commitment to Future Research:

We propose to address this gap in ongoing longitudinal work where expanded socio-demographic profiling is incorporated. We also encourage future replication studies to include these covariates to elucidate their moderating effects.

We thank the reviewer for highlighting this critical issue, which strengthens the transparency of our work. We have added illustration about the control of pre-experimental conditions at the limitation part of the revised manuscript.

  1. Authors should clearly state if the original version of DEBQ-RS, or a Chinese validated one, if so, they have to cite the publication of Chinese validated version.

Response: Thank you for your comments. We have used a Chinese version of DEBQ-RS questionnaire, which was utilized many times in our group and some other groups in China. We have added related references in the revised manuscript.

  1. Authors should state how many publications are related to the same dataset.

Response: Thank you for your comments. This was the first study of the dataset that we prepared. We acquired the dataset from 2024-12-05 to 2025-01-20, then we analyzed the data and write the manuscript. Maybe in the following time, the dataset will be used to compare with other related group such as obesity or lean. We have added illustrations in the revised manuscript.

  1. It is not clear if authors tested the distribution of the sample and checked its normality. This should be reported. As if the sample is not normally distributed, means cannot be used, neither the ANOVA for repeated measures.

Response: Thank you very much for your comments. We have examined the normality of EEG data. As it was 2 (two groups) * 2 (go vs no-go conditions) *2 (high vs low calorie food stimuli) experimental design, 8 variables needed to be verified. From the results, it was found that only the high-go condition in the restrained eater group has not a good normarity, and the other 7 variables were falling into the normality, where they have a mean significant p=0.2834 after the Shapiro-Wilk test. Due to the small sample size, it should be OK to perform the repeated measures-ANOVA.

We have added this in the revised manuscript.

  1. In the Discussion section initially authors should mention their main findings, and extensively report a sort of comparison in concordance and discordance with their finding, as well as the potential reason behind. In the Discussion section in a second place authors should mention the main implication of their finding. Followed by stating clearly and extensively the strength and limitations of their study, foremost the sample size, the external validity. In the Discussion section authors should conclude with the needed new directions for future on the topic.

Response: Thank you for your comments. We have rewritten the discussion part according to your comments.

Finally, thank you for your comments which helps us to improve the quality of manuscript.

Reviewer 2 Report

Comments and Suggestions for Authors

This study presents an investigation into the neural and behavioral mechanisms of memory suppression in the context of food-related cues, particularly in restrained eating.
The relationship between memory inhibition and general inhibitory control could be better delineated. The manuscript appears to conflate general executive functioning deficits observed in obesity with the specific memory suppression processes in normal-weight individuals, without providing a rationale for this extrapolation.
The sample includes only young women, which severely limits generalizability. The rationale for excluding men and older individuals is not discussed. This should be acknowledged as a key limitation.
Forty-two participants is a modest sample for EEG studies, especially when split into subgroups. Was the study sufficiently powered to detect the reported effects, particularly interaction effects?
The statement that "restrained eaters showed a decreasing tendency towards memory control ability" implies causality, which is not justified by the design. A correlational or observational tone would be more appropriate.
The ERP findings are not deeply interpreted. N200 is typically associated with conflict detection and cognitive control — how does this relate to food-specific inhibition or memory suppression in this task?
The distinction between alpha and beta power is presented descriptively but lacks theoretical grounding. What does modulation of these bands mean in the context of memory suppression or restrained eating?
There is inconsistency in terminology (e.g., “restrained eaters” vs. “individuals with restrained eating”).
Phrases like "significant interaction effect" or "group differences" are vague without exact statistical values or effect sizes.
The conclusion that restrained eaters have "impaired inhibitory control over food-related memories" might be too strong given the exploratory nature of the study and its narrow sample.

Author Response

Responds to the comments of Reviewer 2:

Reviewer2:

  1. This study presents an investigation into the neural and behavioral mechanisms of memory suppression in the context of food-related cues, particularly in restrained eating. The relationship between memory inhibition and general inhibitory control could be better delineated. The manuscript appears to conflate general executive functioning deficits observed in obesity with the specific memory suppression processes in normal-weight individuals, without providing a rationale for this extrapolation.

Response: Thank you for your comments. We have rewrite the introduction part according to your comments. We wish the revised manuscript can attain your requirement.

  1. The sample includes only young women, which severely limits generalizability. The rationale for excluding men and older individuals is not discussed. This should be acknowledged as a key limitation.

Response: Thank you for your comments. The reason for the women sample was that most of restrained eaters are belong to female, because women like to restrict their food intake in order to lose weight while men like to use exercise as a weight control technique [Polivy J., Herman C.P., Mills J.S., What is restrained eating and how do we identify it? Appetite 155, 104820, 2020]. The study here was also an extend study of our group [Cui et al., 2023 Appetite, 191, 107048; Bian et al., 2021 Appetite 164, 105269], where they all selected women as participants due to the above reasons.

But for the lack of older individuals, there are several reasons. First, the EEG experiment was a little dear and time-consuming in our institution, therefore, we control the number of participants. Another reason was the memory ability requirement, where the older should be difficult to remember there picture pairs. And the lack of the older participants really reduces the generalization of this study.

We have added the illustration for women participants in the experimental design part, and added the age problem as a limitation in the revised manuscript.

  1. Forty-two participants is a modest sample for EEG studies, especially when split into subgroups. Was the study sufficiently powered to detect the reported effects, particularly interaction effects? The statement that “restrained eaters showed a decreasing tendency towards memory control ability” implies causality, which is not justified by the design. A correlation or observational tone would be more appropriate.

Response: Thank you for your comments.

We have calculated the effect size using ƞ2, which showed a medium or large effect for the relatively small sample. The decreasing tendency of memory control ability should have been overstated. It was resulted from the changes from pre-test to post-test around the TNT modulation, where the restrained eaters showed an increased ACC for high-calorie food cues while the controls showed the expected decreased ACC after memory suppression. Therefore, we stated that the restrained eaters might have poorer memory control ability relative to the controls. As there are only decreased trends and not statically significant, we changed the statement using the observational tone as “an opposing recall trajectories from pre- to post-test for between restrained eaters and controls was observed, where the controls showed the expected decline in recall ACC induced by memory suppression, the restrained eaters exhibited paradoxical enhancement of recall post-test particularly for high-calorie food stimuli.”.

We have added this description in the discussion part of the revised manuscript.

  1. The ERP findings are not deeply interpreted. N200 is typically associated with conflict detection and cognitive control – how does this relate to food-specific inhibition or memory suppression in this task?

Response: Thank you for your comments. We tried to illustrate the association between N200 amplitude and the TNT task as follows: “Higher N200 amplitude for low-calorie food stimuli under think condition than no-think condition might indicate that low-calorie food cues are more difficult to be remembered due to their similar appearance. More conflict was encountered for participants during this period as most of low-calorie food cues are green vegetables with similar appearance, this was also a neglection for our picture selection.” We have added this in the discussion part of the revised manuscript.

  1. The distinction between alpha and beta power is presented descriptively but lacks theoretical grounding. What does modulation of these bands mean in the context of memory suppression or restrained eating?

Response: Thank you for the comment.

Theta and beta rhythms have been studied in obese/overweight individuals with cognitive control tasks, and there are some results for the two rhythms during the cognitive control process. It was also observed by our group with food stimuli in go/no-go tasks [Lian et al., 2025, Cerebral Cortex, 35, bhaf056; Liu et al., 2020, brain topography, 33, 101-111]. However, there are little studies for the restrained eaters’ rhythm oscillations during the cognitive control tasks. Therefore, this study tried to explore if the restrained eaters’ theta/beta oscillation was unique or similar to the control groups. Also, it can have some comparison between obese group and restrained eaters through the normal-weight and not dieting controls. As the two rhythms participated in the TNT task, it should be significant to modulate these neural oscillations from the view of neurofeedback field, by adding or reduce theta/beta oscillations to certain brain region to modulate the participants’ behavioral performance. This was what we want to express.

To make it more concise, we have removed the sentences in the revised manuscript.

  1. There is inconsistency in terminology (e.g., “restrained eaters” vs. “individuals with restrained eating”).

Response: Thank you for your comment. We have corrected this point which was unified as “restrained eaters” in the revised manuscript.

  1. Phrases like “significant interaction effect” or “group differences” are vague without exact statistical values or effect sizes.

Response: Thank you for your suggestions.

We have corrected these descriptions in the revised manuscript.

  1. The conclusion that restrained eaters have “impaired inhibitory control over food-related memories” might be too strong given the exploratory nature of the study and its narrow sample.

Response: Thank you for your comments. We have removed the word “impaired” in the revised manuscript. It was a bit exaggerated and not strict.

Finally, thank you for your comments which helps us to improve the quality of manuscript.

Round 2

Reviewer 1 Report

Comments and Suggestions for Authors

.

Reviewer 2 Report

Comments and Suggestions for Authors

Having seen the manuscript's modifications, I agree with its publication, as it has been improved.